# Helstrom Bound for Squeezed Coherent States in Binary Communication

**DOI:** 10.3390/e24020220

**Published:** 2022-01-31

**Authors:** Evaldo M. F. Curado, Sofiane Faci, Jean-Pierre Gazeau, Diego Noguera

**Affiliations:** 1Centro Brasileiro de Pesquisas Físicas, Rua Xavier Sigaud 150, Rio de Janeiro 22290-180, Brazil; evaldo@cbpf.br (E.M.F.C.); sofiane@cbpf.br (S.F.); diegonoguera.srl@gmail.com (D.N.); 2National Institute of Science and Technology for Complex Systems, Rua Xavier Sigaud 150, Rio de Janeiro 22290-180, Brazil; 3Instituto de Humanidades e Saúde/Departamento de Ciências da Natureza, Universidade Federal Fluminense, Rio das Ostras 28895-532, Brazil; 4CNRS, Astroparticule et Cosmologie, Université de Paris, F-75013 Paris, France

**Keywords:** coherent states, squeezed states, Helstrom bound

## Abstract

In quantum information processing, using a receiver device to differentiate between two non-orthogonal states leads to a quantum error probability. The minimum possible error is known as the Helstrom bound. In this work, we study the conditions for state discrimination using an alphabet of squeezed coherent states and compare them with conditions using the Glauber-Sudarshan, i.e., standard, coherent states.

## 1. Introduction

In quantum information processing, the carrier is a quantum system and the alphabet is encoded in a set of quantum states. To find out which state was sent, the receiver carries out a measurement. When the chosen alphabet is based on non-orthogonal states, there exists a non-zero probability that the receiver might misconstrue the transmitted state [1,2]. Optimized measurement schemes often lead to the so-called Positive Operator Valued Measures (POVMs)). Minimizing the error over all possible POVMs leads to the *quantum error probability*, also known as *Helstrom bound* [3]. This is the smallest physically allowed error probability and defines a criterion for discrimination between transmitted non-orthogonal states. When the sender uses two pure states as codewords, $|\Psi_{0}\rangle |$ and $|\Psi_{1}\rangle$, both having the same prior probability $p_{i}=1/2,i=0,1$, the *Helstrom bound* is given by
(1)$$P_{H}=\frac{1}{2}\left(1-\sqrt{1-|\langle \Psi_{1}|\Psi_{0}\rangle {|}^{2}}\right)\;.$$

Gaussian states constitute the most important example of non-orthogonal states used in quantum communication with optical systems. The most studied gaussian states are the Glauber-Sudarshan (i.e., standard) coherent states (GS-CS) [4], mathematically described as eigenstates of the annihilation operator in Fock space and implemented by lasers operating far above threshold. Note that the GS-CS are the quantum optics version of those special wave packets of harmonic states discovered by Schrödinger in 1926 [5]. Several experimental and theoretical works were published using GS-CS in binary communication [6,7]. Gaussian states produced by squeezed light have also been studied for their use in quantum communication [8,9], as well as many other physical applications like sensitivity enhancement of gravitational waves detectors [10].

Squeezed light exhibits less noise in the electric field at some values of the phase variable than the Fock vacuum state. This is a consequence of the quantum nature of light and cannot be explained within classical physics [11]. More recent reviews on squeezed light can be found in [12,13]. Squeezed states of light were produced for the first time in the mid 1980s, through wave mixing in an optical cavity [14] and by parametric down conversion [15]. Squeezed states can also be produced through photon-adding on coherent states [16].

This work lies in the continuation of a previous exploration [17] of statistical properties and Helstrom Bound for non-standard coherent states. Extending our study to squeezed states is motivated by their relevance on the experimental level, and the importance to find the best parameters in order to minimize the quantum error. In [18], the state discrimination in binary communication with a phase-shift keyed (PSK) alphabet of squeezed coherent states was studied. Here we want to extend the study of state discrimination to alphabets made of two arbitrary squeezed coherent states where the only imposed constraint is fixing the energy (average number of photons) for each letter of the alphabet. Moreover, the Helstrom bound for a pair of squeezed coherent states is compared with its GS-CS counterpart in order to determine the conditions for which state discrimination can be improved. Note that at this stage we do not consider noise nor any kind of losses. One should also mention that other pioneers’ works related to optimal quantum detection and squeezed light—at that time called two-photon coherent states—are found in [19,20,21,22,23,24,25].

The manuscript is organised as follows. In Section 2 we write down the squeezed coherent states and calculate the overlap between two arbitrary such states, the (involved) analytic expression of its modulus squared is detailed in Appendix A. Section 3 discusses the Helstrom bound for several interesting cases, including a comparison with the Glauber-Sudarshan case. Section 4 summarises our conclusions.

## 2. Squeezed Coherent States from Holomorphic Hermite Polynomials

The analytic expression for the expansion coefficients of a single-mode squeezed coherent state in the Fock Basis can be found in [26]. These authors make use of the holomorphic Hermite polynomials to reach a compact form. More details on the relation between squeezed coherent states and the holomorphic Hermite polynomials can be found in [27,28]. See also the seminal paper which introduced the holomorphic Hermite polynomials [29]. For α∈C (CS-GS parameter) and $\xi =re^{i\theta }\;,r\geq 0\;,\theta \in \left[0,2\pi \right)$ (squeezing parameter), the normalised squeezed coherent states |α,ξ〉:=D(α)S(ξ)|0〉 are given in terms of the number states basis by
(2)$$|\alpha ,\xi \rangle =\frac{\exp\left[-\frac{{\left|z\right|}^{2}}{2}\sinh\left(2r\right)+{\left(z^{*}\right)}^{2}\sinh^{2}r\right]}{\sqrt{\cosh r}}\sum_{n=0}^{\infty }\frac{1}{\sqrt{n!}}{\left(e^{i\theta }\frac{\tanh r}{2}\right)}^{\frac{n}{2}}\;H_{n}\left(z^{*}\right)\;|n\rangle \;,$$
where
(3)$$z=\frac{1}{\sqrt{2}}\alpha^{*}e^{\frac{i}{2}\theta }\frac{1}{\sqrt{\tanh r}}+\frac{1}{\sqrt{2}}\alpha e^{-\frac{i}{2}\theta }\sqrt{\tanh r}\;.$$$D\left(\alpha \right)=\exp\left(\alpha a^{†}-\alpha^{*}a\right)$ and $S\left(\xi \right)=\exp\left[\frac{\xi^{*}a^{2}-\xi {\left(a^{†}\right)}^{2}}{2}\right]$ are the unitary displacement (Weyl) and squeezing operators, respectively. These are expressed in terms of the lowering *a* and raising $a^{†}$ operators for (photon) number states |n〉. The $H_{n}\left(\cdot \right)$ are the Hermite polynomials analytically continued to the whole complex plane. Using Mehler’s formula, the states (Equation 2) overlap can be written as
(4)$$\begin{array}{cc} \; & \langle \alpha^{\prime },\xi^{\prime }|\alpha ,\xi \rangle ={\left[\cosh r\cosh r^{\prime }\left(1-e^{i(\theta -\theta^{\prime })}\tanh r\tanh r^{\prime }\right)\right]}^{-1/2}\\ & \times \exp\left[-\frac{|z^{\prime }{|}^{2}}{2}\sinh\left(2r^{\prime }\right)+{\left(z^{\prime }\right)}^{2}\sinh^{2}r^{\prime }-\frac{{\left|z\right|}^{2}}{2}\sinh\left(2r\right)+{\left(z^{*}\right)}^{2}\sinh^{2}r\right]\\ \; & \times \exp\left[\frac{2e^{\frac{i}{2}\left(\theta -\theta^{\prime }\right)}\sqrt{\tanh r\tanh r^{\prime }}z^{\prime }z^{*}-e^{i(\theta -\theta^{\prime })}\tanh r\tanh r^{\prime }\left({\left(z^{\prime }\right)}^{2}+{\left(z^{*}\right)}^{2}\right)}{1-e^{i(\theta -\theta^{\prime })}\tanh r\tanh r^{\prime }}\right]. \end{array}$$

An equivalent expression to this formula can be found in [19]. Note that the squeezed states involved here are labelled by two complex parameters (see Equation (Equation 2)) yet they are elements of a one-mode Fock space spanned by eigenstates |n〉 of the number operator for one-mode photons. They should not be confused with bipartite elements of a two-mode Fock Hilbert space spanned by two-mode eigenstates |m,n〉, see for instance [30,31].

By introducing the parameter $s:=e^{-2r}\in \left(0,1\right]$ and the cartesian coordinates or quadratures (x,y) of z=x+iy one obtains an alternative formulation for the states (Equation 2):(5)$$|\alpha ,\xi \rangle =\frac{\exp\left[\frac{-\left(1-s\right)x^{2}-\left(\frac{1}{s}-1\right)y^{2}-i\left(s+\frac{1}{s}-2\right)xy}{2}\right]}{\sqrt{\frac{\sqrt{s}+\frac{1}{\sqrt{s}}}{2}}}\sum_{n=0}^{\infty }\frac{1}{\sqrt{n!}}{\left(\frac{e^{i\theta }}{2}\frac{\left(1-s\right)}{\left(1+s\right)}\right)}^{\frac{n}{2}}\;H_{n}\left(x-iy\right)\;|n\rangle \;.$$

Finally, it is worth noting that the states (Equation 5) resolve the identity 𝟙 in the Fock Hilbert space of number states,
(6)$$𝟙=\frac{\left(\frac{1}{s}-s\right)}{2\pi }\int_{C}d^{2}z\;|\alpha ,\xi \rangle \langle \alpha ,\xi |\;,\;d^{2}z:=dx\;dy\;.$$

**Remark:** Considering ρ=|α,ξ〉〈α,ξ|, $\alpha^{\prime }=\left(q+ip\right)/\sqrt{2}$ and using the overlap given by (Equation 4), the squeezed coherent states can be represented in optical phase space through the Husimi function $Q\left(\alpha^{\prime }\right)=\langle \alpha^{\prime }\left|\rho \right|\alpha^{\prime }\rangle /\pi ={\left|\langle \alpha^{\prime }|\alpha ,\xi \rangle \right|}^{2}/\pi$.

## 3. Helstrom Bound for Squeezed Coherent States

Let us consider a GS-CS alphabet made of two letters $\left\{\left|\eta \rangle ,\;\right|\eta^{\prime }\rangle \right\}$, having energies, or average number of photons, $\langle \eta |a^{†}a|\eta \rangle ={\left|\eta \right|}^{2}$ and $\langle \eta^{\prime }|a^{†}a|\eta^{\prime }\rangle =|\eta^{\prime }{|}^{2}$, respectively. Let us also set an alphabet of two squeezed coherent states $\left\{\left|\alpha ,\xi \rangle ,\;\right|\alpha^{\prime },\xi^{\prime }\rangle \right\}$ having energies $\langle \alpha ,\xi |a^{†}a|\alpha ,\xi \rangle ={\left|\alpha \right|}^{2}+\sinh^{2}r$ and $\langle \alpha^{\prime },\xi^{\prime }|a^{†}a|\alpha^{\prime },\xi^{\prime }\rangle =|\alpha^{\prime }{|}^{2}+\sinh^{2}r^{\prime }$, respectively. Note that $\sinh^{2}r$ and $\sinh^{2}r^{\prime }$ stand for the energy associated to the squeezing process.

Since we want to compare two different encodings yet going over the same communication channel and the same optical apparatus, it is reasonable to fix the energy of each respective letter in the alphabet:(7)$$N={\left|\eta \right|}^{2}={\left|\alpha \right|}^{2}+\sinh^{2}r\;,\;\;N^{\prime }=|\eta^{\prime }{|}^{2}={\left|\alpha^{\prime }\right|}^{2}+\sinh^{2}r^{\prime }\;.$$

Thus, the mean energy *n* is also fixed. It reads $n=p_{0}N+p_{1}N^{\prime }=\left(N+N^{\prime }\right)/2$ since the average is taken over the probability distribution $\left(p_{0}=1/2,p_{1}=1/2\right)$, i.e., the two states have the same prior classical probability $p_{0}=p_{1}=1/2$. We can write energies *N* and $N^{\prime }$ as a function of the average energy *n* and the ratio ν=N/(2n). The factor 2 allows to have ν∈[0,1]. The setting of two alphabets with the same energy yields ν=1/2.

In a similar way, the parameters $r,r^{\prime }\in R$ can be written as functions of the energies $N,N^{\prime }\in R$ and the ratios $\beta =\sinh^{2}r/N$ and $\beta^{\prime }=\sinh^{2}r^{\prime }/N^{\prime }$. Therefore, we can make the following changes of variables to the overlap (Equation 4):(8)$$\begin{array}{cc} w^{2}= & 2n\nu (1-\beta )\;,\\ {\left(w^{\prime }\right)}^{2}= & 2n\left(1-\nu \right)\left(1-\beta^{\prime }\right)\;,\\ \sinh^{2}r= & \;2n\nu \beta \;,\\ \sinh^{2}r^{\prime }= & \;2n\left(1-\nu \right)\beta^{\prime }\;, \end{array}$$
where we have used $\alpha =we^{i\varphi }\;,w\geq 0\;,\varphi \in \left[0,2\pi \right)$. Using the new variables (Equation 8), the Helstrom bound (Equation 1) can be written as
(9)$$P_{H}\left(n,\nu ,\beta ,\beta^{\prime },\varphi ,\varphi^{\prime },\theta ,\theta^{\prime }\right)=\frac{1}{2}\left(1-\sqrt{1-|\langle \alpha^{\prime },\xi^{\prime }|\alpha ,\xi \rangle {|}^{2}}\right)\;,$$
where the modulus squared of the overlap is given in Appendix A. In what follows we analyse the Helstrom bound for some important cases, including the standard GS-CS case.

### 3.1. Case 1: Glauber-Sudarshan Alphabet

The standard GS-CS are achieved by setting $\beta =\beta^{\prime }=0$ in (Equation 2), which also cancels the dependence on the squeezing phases θ and $\theta^{\prime }$. In this case the Helstrom bound (for two different GS-CS states |α〉 and $|\alpha^{\prime }\rangle$) reads in its most general form as
(10)$$\begin{array}{cc} P_{H}\left(n,\nu ,0,0,\varphi ,\varphi^{\prime },\theta ,\theta^{\prime }\right) & \equiv P_{H}\left(n,\nu ,\varphi ,\varphi^{\prime }\right)\\ & =\frac{1}{2}\left(1-\sqrt{1-e^{-2n+4n\sqrt{\nu (1-\nu )}\cos\left(\varphi -\varphi^{\prime }\right)}}\right) \end{array}$$
and exhibits a minimum value for PSK encoding, i.e., when $\varphi -\varphi^{\prime }=\pi$ and ν=1/2. The latter constraint means equal energy for the two states. This minimum value is used in what follows in order to compare with the genuine squeezed states.

### 3.2. Case 2: Squeezed-Coherent PSK Encoding

Starting with a general squeezed state (Equation 2), phase-shift keying is reached through $\alpha^{\prime }=-\alpha$, i.e., $w=w^{\prime }$, $\varphi -\varphi^{\prime }=\pi$ and $\xi^{\prime }=\xi$, i.e., $\beta^{\prime }=\beta$ and $\theta =\theta^{\prime }$, which also implies ν=1/2 (equal energy). This leads to the most general squeezed states with PSK encoding. In this case the Helstrom bound takes the form
(11)$$\begin{array}{cc} P_{H}(n,1/2,\beta ,\beta ,\varphi , & \varphi +\pi ,\theta ,\theta )=\\ \frac{1}{2} & \left(1-\sqrt{1-e^{-4n\left(1-\beta \right)\left(1+2n\beta +2\cos\left(2\varphi -\theta \right)\sqrt{n\beta \left(1+n\beta \right)}\right)}}\right). \end{array}$$

We learned from one of the reviewers that this result, yet in a different form, can be found in [23]. The reviewer also showed the equivalence since Ref. [23] uses different variables. Back then, squeezed light was called two-photon coherent state.

### 3.3. Case 3: Optimal Squeezed-Coherent PSK Encoding

The optimal squeezed coherent state with PSK encoding corresponds to φ=θ/2 and
(12)$$\beta =\beta_{PSK_{opt}}=n/\left(1+2n\right).$$

Note that this optimality condition does not hold for a general squeezed state. This is equivalent to the squeezed states studied in [18], even though the authors had fixed φ=θ=0 from the start. This means that their expression is actually valid for more general values of θ and φ as we have shown here. Please see the next case and Figure 1 for further details.

Figure 1 shows the Helstrom bound for the above three cases for ν=0.5 (same energy in the two channels), $\theta =\theta^{\prime }=0$, φ=0 and $\varphi^{\prime }=\pi$ (PSK encoding). The curve of squeezed *Case 2* for $\beta =\beta^{\prime }=1/2$ appears below that of the standard GS-CS *Case 1* for n≳0.25 while the optimal squeezed *Case 3* stands below the other two curves for any value of *n*. This means that squeezing improves state discrimination, especially for the optimal case.

### 3.4. Case 4: General Squeezed-Coherent Alphabet

For a general Squeezed-Coherent alphabet $\left\{\left|\alpha ,\xi \rangle ,\;\right|\alpha^{\prime },\xi^{\prime }\rangle \right\}$ it is possible to have different squeezing parameters $\beta \neq \beta^{\prime }$ and/or different energies ν≠1/2. The latter condition means there is more energy allocated into one state at the expense of the second of each alphabet.

We have numerically analysed the different possibilities and found that the lowest possible Helstrom Bound case corresponds to the squeezed states with PSK encoding given in *Case 3*. This means that there is always a region (of parameters) where the Helstrom bound is lower for the squeezed states with PSK encoding.

For example, relaxing the constraint ν=1/2 while maintaining $\beta =\beta^{\prime }$ leads to Helstrom bounds with similar behaviour in terms of the angle parameters, yet with higher values than the optimal squeezed PSK case, see Figure 2.

Another example is given in Figure 3, representing an alphabet made of general squeezed coherent states |α,ξ〉 with n=1, $\beta =\beta^{\prime }$, φ=0, $\varphi^{\prime }=\pi$, $\theta^{\prime }=0$ and ν=0.7 in comparison with a Glauber-Sudarshan alphabet. In this case we can see that state discrimination is more efficient for the squeezed coherent state alphabet around θ=0 and for β in some domain. Consequently, dealing with channels with different energies it is still advantageous to use squeezing to improve state discrimination, as is the case for squeezed state with PSK encoding. Note that the optimal β is slightly smaller than 1/3 and thus does not coincide with the optimal squeezed PSK value given by (Equation 12).

To further explore this point we plot the Helstrom Bound as function of $\beta =\beta^{\prime }$ for different values of ν. All other parameters are fixed, the angles to the optimal squeezed PSK values and the mean energy is set to n=1, see Figure 4. It is clear that the optimal β (giving the lowest Helstrom bound for each case) depends on ν and thus is not given by (Equation 12) in the general case. Moreover it is also clear that the optimal squeezed PSK states (i.e., for ν=1/2) do better than all other states.

Finally, relaxing the constraint $\beta =\beta^{\prime }$ shows an interesting feature of the general squeezed states. The squeezed PSK states exhibit a Helstrom bound which is symmetric with respect to β and $\beta^{\prime }$, with lower values for $\beta =\beta^{\prime }$, becoming optimal for $\beta_{PSK_{opt}}$. While the Helstrom bound for general squeezed states is asymmetric and exhibits lower values for $\beta \neq \beta^{\prime }$. In fact all cases are symmetric under the transformation $\nu \to \nu^{\prime }=1-\nu ,\beta ↔\beta^{\prime }$. Figure 5 and Figure 6 show squeezed PSK (ν=1/2) along with genuine squeezed states having ν=1/5 and ν=4/5, respectively. The symmetry between the two cases is clearly shown comparing the two figures.

Note that the squeezed PSK states exhibit lower quantum error overall, but there are still regions where the general case outperform the squeezed PSK case. This can be seen in the right corner of Figure 5 and left right corner of Figure 6, for instance. Nonetheless those regions do not lead to the lowest Helstrom limit within the space of parameters. The lowest possible Helstrom bound corresponds to the optimal squeezed coherent states with PSK encoding, i.e., ν=1/2 and $\beta =\beta^{\prime }=\beta_{PSK_{opt}}$ (Equation 12).

## 4. Conclusions

This work has to be viewed as a natural continuation of a previous exploration [17] of statistical properties and Helstrom Bound for non-standard coherent states. The extension of our previous study to squeezed states is motivated by their relevance on the experimental level, and the importance of finding the best parameters in order to minimize the quantum error.

With the analytic expression for the expansion coefficients of a single-mode squeezed coherent state in the Fock basis, and written in terms of the holomorphic Hermite polynomials, a simple expression for the overlap of two arbitrary squeezed coherent states has been found using Mehler’s formula. The modulus squared of this expression is provided in Appendix A for completeness.

To compare two different encodings, a restriction was imposed; each letter of the alphabet gets a fixed energy. We have then performed a numerical analysis and have shown that, for an alphabet of squeezed coherent states, one can always find regions where state discrimination is improved in comparison to the GS-CS alphabet. The optimal situation corresponds to squeezed coherent states with PSK encoding. This alphabet was studied in [18,23] and is further explored here. What is more, we have shown that this case is valid for more general values of the angular parameters θ and φ, provided φ=θ/2.

We have also shown that the Helstrom bound for general squeezed states is asymmetric with respect to the squeezing parameters β and $\beta^{\prime }$. In this case, the values of β and $\beta^{\prime }$ lowering the Helstrom bound depend on the value of ν and thus are not given by (Equation 12). Moreover, there are regions where the general case outperforms the squeezed PSK case.

The squeezed states studied in this work are labelled by two complex parameters (see Equation (Equation 2)) yet are elements of a one-mode Fock space. The extension to two- and multi-mode Fock Hilbert space might be considered in a future work. Finally, the uncertainty relation of squeezed states and associated correlations of electric and magnetic fields might also be interesting, and will be studied elsewhere.

## Figures and Tables

**Figure 1 entropy-24-00220-f001:**
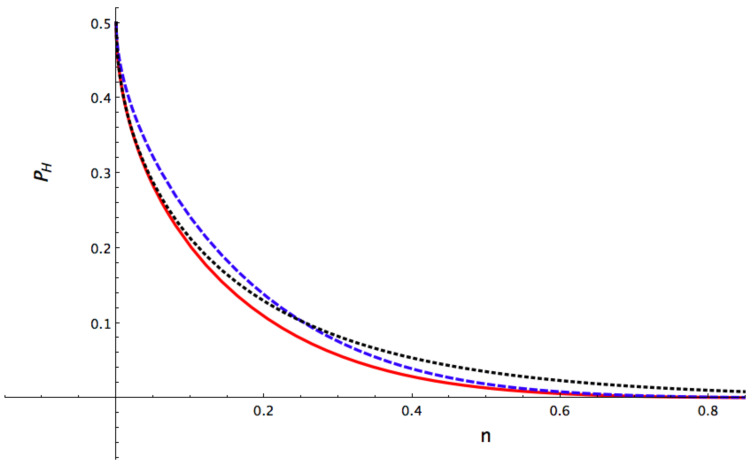
Helstrom bound for ν=0.5, φ=0, $\varphi^{\prime }=\pi$ and $\theta =\theta^{\prime }=0$ using different coherent states with PSK encoding as a function of the mean energy *n*. The black dotted curve represents the Glauber-Sudarshan coherent states treated in *Case 1*. The blue dashed curve depicts the squeezed *Case 2* with $\beta =\beta^{\prime }=0.5$. The red curve pictures the optimal *Case 3*, i.e., with $\beta =\beta^{\prime }=\beta_{PSK_{opt}}$ (Equation 12). Notice that the curve of squeezed *Case 2* is below that of GS-CS *Case 1* for n≳0.25 while the optimal *Case 3* stands below the other two curves for any value of *n*.

**Figure 2 entropy-24-00220-f002:**
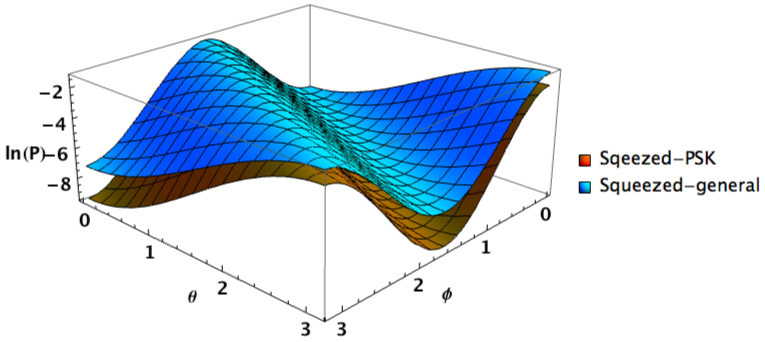
Helstrom bound as function of φ and θ, for n=1, $\beta =\beta^{\prime }=1/3$, (which corresponds to the optimal case of squeezed PSK) $\varphi^{\prime }=\varphi +\pi$, and $\theta^{\prime }=\theta$. The brown-orange surface depicts the ν=0.5 squeezed PSK case while the light blue surface pictures a general squeezed case with ν=0.1. One can easily see that the minimum value for Helstrom bound corresponds to 2φ−θ=0.

**Figure 3 entropy-24-00220-f003:**
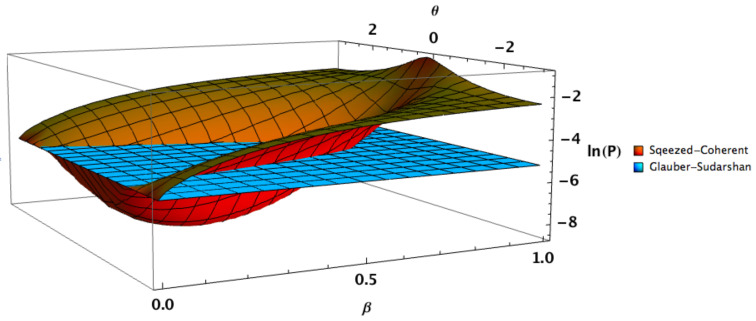
Helstrom bound for n=1, ν=0.7, $\varphi =\theta^{\prime }=0$, $\varphi^{\prime }=\pi$. Blue flat surface: Glauber-Sudarshan alphabet. Brown-orange curved surface: squeezed coherent state alphabet with $\beta =\beta^{\prime }$.

**Figure 4 entropy-24-00220-f004:**
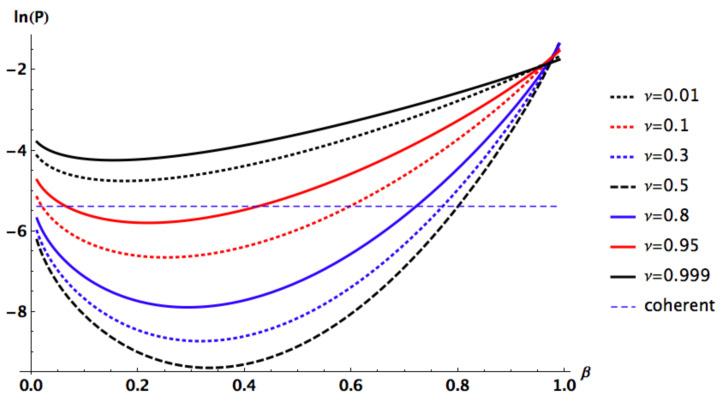
Helstrom bound as function of $\beta =\beta^{\prime }$ for different values of ν while n=1, $\varphi =\theta =\theta^{\prime }=0$, $\varphi^{\prime }=\pi$. The Glauber-Sudarshan case ($\beta =\beta^{\prime }=0$) for ν=1/2 is also plotted for comparison. The optimal β depends on ν and thus is not given by (Equation 12) in the general case, since for $\beta =\beta^{\prime }>4/5$ the Glauber-Sudarshan coherent states present a lower Helstrom bound than the PSK case. The optimal squeezed PSK states (ν=1/2) yields the lowest possible Helstrom bound ($\beta =\beta^{\prime }=1/3$ for n=1).

**Figure 5 entropy-24-00220-f005:**
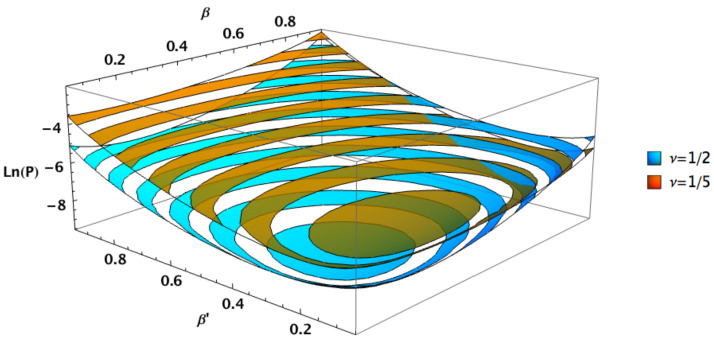
Helstrom bounds for two squeezed states distinguished by their energy channel shares (ν=1/2 and ν=1/5) as functions of arbitrary squeezing parameters $\beta ,\beta^{\prime }$, with n=1, $\varphi =\theta =\theta^{\prime }=0$ and $\varphi^{\prime }=\pi$.

**Figure 6 entropy-24-00220-f006:**
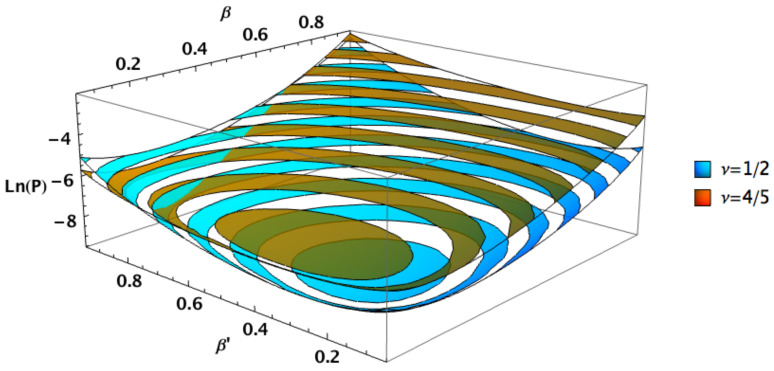
Helstrom bounds for two squeezed states distinguished by their energy channel shares (ν=1/2 and ν=4/5) as functions of arbitrary squeezing parameters $\beta ,\beta^{\prime }$, with n=1, $\varphi =\theta =\theta^{\prime }=0$ and $\varphi^{\prime }=\pi$.

## Data Availability

Not applicable.

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
