# Peer review of "Helstrom Bound for Squeezed Coherent States in Binary Communication"

_entropy, 2022, doi:10.3390/e24020220_

Round 1
Reviewer 1 Report
The paper “Helstrom Bound for squeezed coherent states in binary communication ” and written by Evaldo M. F. Curado et al. presented to me for review is very interesting. The authors build the Glauber-Sudarshan coherent states by using the holomorphic Hermite polynomials and calculate the quantum error probability known as the Helstrom bound which seems to be natural procedure.
As I understand their coherent states are the multiplication of two holomorphic Hermite coherent states. Let us say |z_1, z_2> = sum_{n, m} H_n(z_1) H_m(z_2) |n, m>. But appeared the question: do the authors try to calculate Helstrom bound for the coherent states |z_1, z_2> = sum_{n, m} H_{n, m} (z_1, z_2) |n, m>?
The bipartite holomorphic polynomials H_{n, m} (z_1, z_2) described in
[1] K. Górska, A. Horzela, F. H. Szafraniec, Coherence, Squeezing and Entanglement: An Example of Peaceful Coexistence, DOI: 10.1007/978-3-319-76732-1_5
[2] K. Górska, A. Horzela, F. H. Szafroniec, Holomorphic Hermite polynomials in two variables, J. Math. Anal. Appl. 470 (2019), https://doi.org/10.1016/j.jmaa.2018.10.024
The authors should remember of seminal paper which introduce the holomorphic Hermite polynomials: S.J.L. van Eijndhoven, J.L.H. Meyers, New orthogonality relations for the Hermite polynomials and related Hilbert spaces, J. Math. Anal. Appl. 146 (1990) 89–98.
Author Response
OUR ANSWER TO REFEREE I
We thank the Reviewer for positive assessment and suggestions. Following Reviewer’s recommendations, we added the three references. However we would like to amend a point raised by the Referee. In our paper the involved squeezed states are labelled by two complex parameters (see Eq. 2) but they are elements of a one-mode Fock Hilbert space spanned by eigenstates In> of the number operator for one-mode photons. They are not element of a two-mode Fock Hilbert space spanned by eigenstates Im,n>.
All relevant changes in the manuscript are marked in red.
Reviewer 2 Report
New results can be found in Figs 2 and 3.
But, Introduction, Conclusion, and References are poor. Please do not skip pioneers' works related to the optimal quantum detection and its application to squeezed light signals. I hope the authors include their own view about the following works to this paper.
[1] Osamu Hirota, "Generalized Quantum Measurement Theory and Its Application in Quantum Communication Theory (Optical Communication by Two-Photon Laser)," Electronics and Communications in Japan 60-A (8), 701-708 (1977) (in Japanese); (English translation by Scripta Publishing Co., 1978)
[2] Osamu Hirota, Isamu Shioya, Shikao Ikehara, and Yasuharu Suematsu, "Optimal Control of Quantum Noise (Optical Communication by Two-Photon Laser Part II)," Trans. IECE Japan E-61 (4), pp. 273-279 (1978).
[3] Horace P. Yuen and Jeffrey H. Shapiro, "Optical Communication with Two-Photon Coherent States --- Part I: Quantum-State Propagation and Quantum Noise Reduction," IEEE Trans. Inf. Theory IT-24 (6), 657 (1978).
[4] Jeffrey H. Shapiro and Horace P. Yuen, "Optical Communication with Two-Photon Coherent States --- Part II: Photoemissive Detection and Structured Receiver Performance," IEEE Trans. Inf. Theory, IT-25 (2), 179 (1979).
[5] Horace P. Yuen and Jeffrey H. Shapiro, "Optical Communication with Two-Photon Coherent States --- Part III: Quantum Measurements Realizable with Photoemissice Detectors," IEEE Trans. Inf. Theory, IT-26 (1), 78 (1980).
[6] Osamu Hirota (Ed.), Squeezed light (Elsevier, Amsterdam, 1992).
In particular, I strongly recommend the authors to check Eq.(19) of [1], Eq.(10) of [2], Eqs. (3.41) and (4.45) of [4].
Author Response
We thank the Reviewer for positive assessment and suggestions. We agree about criticisms and suggestions. So we have improved our introduction and conclusion. We have also added the six references recommended by the Reviewer by quoting them at the appropriate place.
All relevant changes in the manuscript are marked in red.
Reviewer 3 Report
The MS aims at studying how Helstrom quantum detection theory (developed in the 1970s by Holevo and Helstrom) can be applied to quantum communications using squeezed optical states. I found the MS neither substantial nor convincingly original.
First I will start with nomenclature. The authors strangely use the terminology “Glauber-Sudarshan coherent states.” This is very misleading and historically wrong. Coherent states were discovered by Erwin Schrodinger in the 1920s shortly after he discovered the equation bearing his name. The subject was important since the birth of quantum theory because coherent states were derived by Shrodinger in his attempt to find the most classical quantum state possible (solution to driven harmonic oscillators). On the other hand, Sudarshan discovered the P-representation of coherent states in the early 1963, and Glauber further developed the same idea and subject in the same year. (The P-representation is sometimes called the Glauber-Sudarshan representation.)
I am finding it strange that one of the authors wrote a book on the subject of coherent states, Ref [5], and yet the historical facts are not right in this MS.
Also, the authors mention that coherent states are used in quantum communications (correct), but then cite references like [4] and [5], which are both quite recent (21st century). In fact, optical communications using coherent and other quantum states was discussed as early as the 1960s and 1970s. For example, Helstrom discusses extensive physical apparatus in his book cited here as [3]. So the authors give the misleading impression that this subject is emerging.
The same can be said about squeezed states. This subject has been exploding for nearly three decades. I don’t understand why it is being presented as a new subject (quantum communications using squeezed states.)
Now we come to the most serious problem: lack of originality. The MS is incremental. The key “idea” appears to perform parametric study of a quite straightforward and well known textbook closed-form analytical expression. Formula (4), which is a special example of Helstrom probability of error of binary quantum shift keying, equation (1) in the MS, is well known and can be found in all previous references on squeezed states as the authors surely know. However, what I don’t understand is what exactly is the point of plugging some numbers and looking into whether some values are “optimal” or not. This is a very straightforward exercise and none of the conclusions that follow seem even mildly interesting (in my opinion).
To be more specific, I can write in just 10 minute a Python or MATLAB script to compute the inner product of two squeezed states given in (4) in the MS. There is absolutely no need for the Appendix, which is clearly produced using a symbolic computer tool and is neither interesting nor even needed for the results in the paper. The entire “results” of this paper could be produced by simple parametric study of this 10-min-to-write computer code to implement a textbook formula.
In other words, once you plug the four real parameters of zeta (complex) and alpha (complex) into (4), you can obtain any combination you need for evaluating the Helstrom bound of a binary quantum communication scheme. There is no noise analysis here. No actual quantum evolution analyzed. No solution of new equations or alternative ideas. The MS does not actually establish anything new except commenting on some trivial parametric study of computing (4).
In my opinion, the paper is more like a routine conference paper or a short note. It does not qualify as a journal paper.
Author Response
OUR ANSWER TO REFEREE III
We wrote our paper as a needed complement to the recently published paper by us (Ref 9, former version, J. Opt. Soc. Am. B 2021, 38, 3556–3566), in which we examined the question of the Helstrom bound for various families of coherent states. The fact we extend our study to squeezed states is motivated by their relevance on the experimental level, and the importance to find the best parameters in order to minimize the Helstrom bound, enhancing the communication channels. In this sense it might be viewed as « incremental ». Actually, it is incremental, as are most of the papers. The word "incremental" means that a small improvement was made, otherwise there is no incremental. In fact, in figs. 2 and 3 we present the best options to prepare the squeezed states in communication. The important question is wether or not the content of our work brings new information with regard to the huge literature devoted to the subject, and in particular to our previous paper. In the new version, we insist on this point in the introduction and in the conclusion.
Concerning the point raised by the Referee about the modulus squared of the overlap of two squeezed states, we agree that those analytical formulae appear as intricate. However an explicit analytical expression is always important and deserves to be published. It allows to see hidden symmetries and other important aspects for the purpose of the paper. Our expression was obtained both by hand and by software. In fact, the best way to construct the squeezed coherent states in such a way to minimize the Helstrom bound was found through a careful analysis of the analytical expression.
Concerning the nomenclature, we agree with Referee’s assertion as which « The P-representation is sometimes called the Glauber-Sudarshan representation. ». Now, we know and acknowledge the fundamental contribution of Schroedinger (Der steatite Obergang von der Mikro-Zur Makromechanik, Die Naturwissenshaften, 1926) to the long history of « coherent » states, pointed out in the preface and in the introduction of the book written by one of us. But we would like to respectfully remind the Reviewer that our work pertains to quantum optics and not to quantum mechanics, and that the adjective « coherent » was proposed by Glauber (PRL 10, 1963) within the context of quantum electrodynamics or quantum optics. The name of Sudarshan is added for obvious and fair reasons. Those states are also called canonical coherent states (CCS), standard coherent states, Gaussian states, or oscillator states, or even Klauder-Glauber-Sudarshan.
Concerning Schroedinger, it is valuable to quote here his own motivations:
« I would like to show here the transition to macroscopic mechanics by simple and concrete examples, in which I show that the packet of eigenmodes with high n and with relatively small “difference in quantum numbers” may represent the mass point that moves according the usual classical mechanics, i.e. it oscillates with frequency ν0 ».
We have also corrected some minor typos, as mentioned by the reviewer.
All relevant changes in the manuscript are marked in red.
Round 2
Reviewer 2 Report
Please check my comments in the attached PDF.

Author Response
We thank the Reviewer for a careful and detailed examination of our article and for useful comments and suggestions. The latter helped us to significantly improve its content. All changes with respect to the previous version are highlighted in red. Below are our answers to Reviewer comments/remarks.
- Reviewer
The submitted paper concerns “communication” as indicated in the title. Explain why ”the total energy” is defined and used, or define the average and replace all to the average. It is natural to define the average, not the total, in the theory of communication because two signal states never co-exist at the same time. The receiver receives only one of the two signals in each time.
Our answer
The Reviewer is right. We have mentioned in the introduction of the new version that p0 = p1 = 1/2 (before Eq. (2)). In section 3, we have corrected the expression of the average energy n = p0 N + p1 N′ after eq. (7), which leads to 2n = N + N′.
Therefore, we perform the replacement n ---> 2n over all the manuscript.
2. Reviewer
Check and correct Eq. (11). Refer the paper [Yuen(1976)] for Eq. (11).
• Eq. (2) is equivalent to Eq.(3.33) of Yuen PRA (1976), 13:6, 2226.
• Eq. (11) might be incorrect.
• Eq. (11) is not a new. An equivalent and pretty simple form can be found in Eq.(3.45) of Ref. [Shapiro(1979)]. Please do not ignore Shapiro-Yuen’s result for binary PSK squeezed state signals.
Our answer
The Reviewer is right
• Eq. (2) is indeed equivalent to Eq.(3.33) of [Yuen(1976)]. We mention this in the new version of the manuscript.
• Eq. (11) has been corrected in the new version. We also cite and give credit to [Shapiro(1979)].
3. Reviewer
Original work is only in Subsection 3.4. Extent this part.
Our answer
We have greatly enhanced this subsection, adding new figures and further analysis.
Reviewer 3 Report
Thank you for your response. You answered all my questions. I am very happy with the quote from Schrodinger in particular.
By the way, I might respectfully disagree with the authors about the following issues:
1) The fact that Glauber used the term "coherent" (provided he was indeed the first one) does not mean he discovered these states. Please check the extensive book by Klauder and Sudarshan on quantum optics published in 1965 which lists papers in mathematical physics studying coherent states in the 1950s. In fact, Sudarshan developed his fundamental P-representation even before Glauber but Glauber's name made the topic famous. However, this is not a historical paper so I will not raise the issue again.
2) Regarding the quality of the contribution, I expressed my opinion that more work is needed to make this a substantial journal paper. I appreciate the honesty and work of the authors, but in my personal view I usually prefer to see more content in a journal paper than this. Maybe Entropy is a different journal than what I am used to. The AE will decide whether this MS is ok for this journal or not.
3) While I agree with the authors on the importance of the subject of applying Helstrom theory to squeezed states, I personally did not see that the lengthy expressions in the Appendix are even mildly interesting (even though I am a theorist). I don't see anything new in them. They are not essential for the results since you may just produce all of these curves using a short python or MATLAB script.
Author Response
We thank the Reviewer for positive assessments and further comments/remarks.
Concerning the second paragraph of Reviewers report concerning the quality of the contribution, we have significantly extended its content, particularly in Subsection 3.4. All changes with respect to the previous version are highlighted in red in the supplementary document.
Round 3
Reviewer 2 Report
The Helstrom bound is a milestone of quantum information science. The squeezed light too. This work will recall the significance of them again to the readers, together with that of the pioneering works.